# Face Mask Detection on Photo and Real-Time Video Images Using Caffe-MobileNetV2 Transfer Learning

**B. Anil Kumar and Mohan Bansal ***

School of Electronics Engineering, VIT-AP University, Amaravati 522237, AP, India
* Correspondence: mohan.bansal@vitap.ac.in

**Abstract:** Face detection systems have generally been used primarily for non-masked faces, which include relevant facial characteristics such as the ears, chin, lips, nose, and eyes. Masks are necessary to cover faces in many situations, such as pandemics, crime scenes, medical settings, high pollution, and laboratories. The COVID-19 epidemic has increased the requirement for people to use protective face masks in public places. Analysis of face detection technology is crucial with blocked faces, which typically have visibility only in the periocular area and above. This paper aims to implement a model on complex data, i.e., by taking tasks for the face detection of people from the photo and in real-time video images with and without a mask. This task is implemented based on the features around their eyes, ears, nose, and forehead by using the original masked and unmasked images to form a baseline for face detection. The idea of performing such a task is by using the Caffe-MobileNetV2 (CMNV2) model for feature extraction and masked image classification. The convolutional architecture for the fast feature embedding Caffe model is used as a face detector, and the MobileNetV2 is used for mask identification. In this work, five different layers are added to the pre-trained MobileNetV2 architecture for better classification accuracy with fewer training parameters for the given data for face mask detection. Experimental results revealed that the proposed methodology performed well, with an accuracy of 99.64% on photo images and good accuracy on real-time video images. Other metrics show that the model outperforms previous models with a precision of 100%, recall of 99.28%, f1-score of 99.64%, and an error rate of 0.36%. Face mask detection was originally a form of computing application, but it is now widely used in other technological areas such as smartphones and artificial intelligence. Computer-based masked-face detection belongs in the category of biometrics, since it includes using a person's unique features to identify them with a mask on.

**Keywords:** deep neural network; face detection; face mask classification; Caffe-MobileNetV2 model; transfer learning

## 1. Introduction

The face detection system is one of the most widely used techniques for detecting human faces in digital photos and video images. Object classification in image processing aims to detect all different shapes and colours of objects present in images and classify them into the appropriate classes. Face detection, emotion classification, gender identification, and face mask recognition are examples of image processing tasks. The image processing technique describes the process of extracting valuable information from images by applying procedures to them. Importing an image, performing operations, and analysing the input image are some of the stages involved in producing the final results of the transformed images.

Face detection has become an important topic due to its potential applications in biometrics, human–computer interaction (HCI), and surveillance. It is critical to supervise and monitor the use of face masks in public places. Despite advances in generic target detection algorithms in other disciplines, face mask detection techniques are currently less effective [1]. Some research has been conducted in this area by researchers who used

the "you only look once v2" (YOLOv2) algorithm to build the detection model [2]. The YOLOv3 algorithm is also utilized to improve feature extraction, which is an improved spatial pyramid-like pooling structure [3]. Face mask detection also employs an algorithm for eyelid face detection and an improved self-attention mechanism of the feature pyramid network [4]. A modified YOLOv4 algorithm was developed for a tiny lightweight network technique to obtain the most relevant features of multiple targets [5]. In general, the YOLO network creates $N$ predictions for bounding boxes for every grid with size $G \times G$ from an image. The network cannot discover smaller items since each bounding box can only have one class during prediction. Since the ratio of the bounding box is completely learned from data, the major cause of YOLO's problem is localization. This is due to YOLO's errors with unusual ratio bounding boxes [6].

Several transfer learning approaches can also be used to solve the problem of face mask detection in the real world. As a transfer learning method, a pre-trained InceptionV3 model is used to detect people with or without masks [7]. The hybrid deep transfer learning model that combines deep learning methods with traditional computer learning is used for face mask detection [8]. The model was also created with a transfer learning concept developed on the pre-trained MobileNetV2 model for real-time face mask detection and localization [9].

Convolutional neural networks (CNN) are used in a cascading approach for detecting masked faces [10]. In terms of modern approaches, the retina face mask network is designed as a unique framework for correctly and effectively recognizing face masks [11]. Several experiments [12–14] were conducted to develop a technique that can automatically determine whether or not the person is wearing a face mask. The YOLOv3 technique and the haar cascading classifier used to detect facial masks in real-time [15]. A comparison of four reducing deep learning approaches are used in face recognition, namely, VGGFace, FaceNet, OpenFace, and DeepFace [16]. A visual geometry group-16 (VGG-16) is also used in deep neural network (DNN) methods [17] for face mask detection. The VGG-16 architecture for real-time face mask recognition is used to create real-time facial mask identification with an alert system [18].

One of the well-researched practical issues is face detection and recognition. Significant progress has been made in facial detection technology in last few years. It is difficult because of the variations in facial structure and the existence of masks. The main goal of this research is to improve the detection performance of various masked and unmasked faces, particularly with regard to facial masks. The challenge of face mask detection in the field of image processing and in computer vision has been shown to be extraordinary. The main focus of the proposed research is to improve human protection by utilizing a deep learning platform to identify people wearing a mask or not in public areas. The mask detector will certainly and properly be utilized to help secure our protection. Furthermore, it is sad to have lived during the COVID-19 period and witnessed what happened in the world at the time. This motivates us to transform a real-world problem of having a habit of wearing masks when going outside using machine learning techniques.

The face mask detection method proposed in this work used a transfer learning approach to a pre-trained MobileNetV2 model, and the Caffe model as a single-shot multibox detector for fine-tuning. Face mask detection is a natural extension of face detection in the field of computer vision. The Caffe model works as a face coordinates extractor; the classifier predicts the outcomes based on confidence values after receiving the face dimensions. It is an easy task for humans to detect whether someone is wearing a face mask or not by visualizing a human face, but it is more difficult for machines. This problem is described as a computer vision issue that can be solved with the help of deep learning (DL) techniques. This technology has the potential to be applied in a wide range of real-world applications, including cancer detection, coronavirus identification, X-ray sample analysis, crime scenes and for security improvement.

Previous studies and existing research methods have not focused on the problem of developing a single model for detecting both photo and real-time video images with

and without masks. All research studies have been performed on real-time video or photo images separately. The main goal of our research is to create a model that can solve these problems with lesser parameters, low computation cost, fast computation and better accuracy.

The proposed Caffe-MobileNetV2 (CMNV2) model was used as a transfer learning approach for face mask detection. The experiments were performed on the image dataset and the webcam video images. In this work, OpenCV was used to detect human faces and a deep learning technique was used to recognize the region of interest (ROI) as that of the person's face. The classification was performed using lightweights such as MobileNetV2 and Caffe models such as feature extractors and image classifiers. The proposed CMNV2 model was trained on the two-class image dataset and tested on the image dataset and the webcam video images for the face mask classification. The bounding boxes were drawn around the faces with either green or red colours based on the output, along with the classified class name. The main contribution of the work is:

i.  To train and test the CMNV2 model for detecting face masks in the photo image dataset and real-time video images.
ii.  A transfer learning method was used by adding five of our own layers in the MobileNetV2 model and developeing a modified architecture (CMNV2) as the proposed model.
iii.  Additionally, a modified MobileNetV2 model was used for various image dimensions, including $224 \times 224$, $192 \times 192$, $160 \times 160$, and $128 \times 128$ to measure the performance of the proposed model with different image dimensions.
iv.  To select the best model, the modified MobileNetV2 model with different image dimensions was compared to the existing MobileNetV1 model with different image dimensions along with the Caffe model.

The remainder of the paper is categorized as follows: The related work is described in Section 2. Section 3 discusses the significance of standard convolution and the convolution used in the modified MobileNetV2. The proposed methodology using the modified MobileNetV2 with Caffe model for face detection is described in Section 4. The performance of the proposed methodology and the confusion matrix and various metrics are described in Section 5. The results containing face detection with and without masks along with the proposed model accuracy are mentioned in Section 6. Finally, the paper's conclusion is summarised in Section 7.

## 2. Related Work

Several recent studies have been performed on face mask detection during COVID-19, and those are presented in the literature. Deep learning-based approaches were developed by researchers to study the issue of face mask detection [19]. To address the issue of masked face recognition, ResNet50 was developed as a reliable solution based on occlusion removal, as well as deep learning-based features [20]. AlexNet and VGG16 convolutional neural network designs are used as transfer learning for the development of new models [21]. Technology has been developed that prevents the spread of viruses and uses deep learning technology to ensure that people are wearing face masks correctly. The CelebA dataset was used to develop a model to automatically remove mask objects from the face and synthesize the corrupt regions while maintaining the original face structure [22]. A multi-threading strategy with VGG-16 and triplet loss FaceNet dependent on the masked faced recognition approach is proposed, which is built on MobileNet and Haar-cascade to detect face masks [23]. The embedding unmasking model (EUM) method was designed, which aimed to improve upon existing facial recognition models. The self-restrained triplet (SRT) technique was utilized, which allowed EUM to produce embeddings corresponding to the associated characters of unmasked faces [24]. The margin cosine loss (MFCosface) masked faced recognition algorithm, which is dependent on a wide margin cosine loss design, was proposed for detecting and identifying criminals. An attention-aware mechanism was improved by incorporating important facial features that helps in recognition [25].



An attention-based component using the convolutional block attention module (CBAM) model was designed, which depends on the highlighted area around the eyes [26]. The near-infrared to visible (NIR-VIS) masked faced recognition problem was analyzed in terms of the training approach and data model [27]. A method called heterogeneous semi-siamese training (HSST) was designed, which attempts to maximize the joint information between face representations using semi-siamese networks.

A real-world masked face recognition dataset (RMFRD) is the largest real-world masked face dataset used for research on the face verification problem by researchers. An identity-aware mask generative adversarial network (IAMGAN) using a segmentation-guided multi-level identification module was used to produce the artificial masked face images from full face images [28]. A methodology utilized its own FaceMaskNet-21 deep learning model and also the deep metric learning approach to generate 128-dimension (128-d) face encodings [29]. This helps in face identification in static images and live video streams, as well as static video files. An attention-based approach for recognizing masked faces was proposed by combining a cropping-based strategy with a convolutional block attention module (CBAM) [30]. The attention-based model, diverse and sparse attentions-Face (DSA-Face), was developed for face recognition and face matching [31]. The DSA-Face model is composed of pairwise self-contrastive attentions (PSCA) and a sparsity loss (ASL). The PSCA was able to extract diverse local representations by enlarging pairwise attention distances, whereas the ASL shrinks responses from distracted regions in attention maps towards zero.

The single shot multibox MobileNetV2 (SSDMNV2) approach uses the single shot multibox detector as a face detector, and the MobileNetV2 design serves as the classifier [32]. An end-to-end de-occlusion distillation architecture containing two modules has been developed as a methodology to transfer the mechanism of a model completion for the problem of masked face recognition [33]. The multi-granularity (MGL) model was designed for three different forms of masked face datasets for recognition, such as the simulated masked face recognition dataset (SMFRD), the masked face detection dataset (MFDD) and the real-world masked face recognition dataset (RMFRD) [34]. A novel latent part detection (LPD) approach was used to find the latent face portion that is robust to mask use and it was then utilized to extract discriminative features [35]. A face mask recognition approach that distinguishes between images with and without masks has been developed for static images and real-time videos [36]. The ensemble of the first two-stage detectors was developed for low inference time and high accuracy. A bounding box transformation was also developed to increase the effectiveness of localization during mask recognition [37].

Deep neural networks can be used to handle image detection and recognition problems because the problem has grown more complex over time. However, adding more layers to the networks makes them more complex and challenging to train; as a result, accuracy degradation is frequently observed. ResNet, which stacks additional layers and achieves improved performance and accuracy, was developed to address this problem [38]. Complex characteristics can be learned by the additional layers; however, the optimal number of layers to add must be determined experimentally in order to prevent any model performance reduction. One of the most significant and widely applied lightweight deep neural networks for face recognition tasks is MobileNet [39], which is primarily dependent on a streamlined design. Its architecture displayed a strong hyperparameter performance, and faster calculations of the model [40]. As another well-known CNN-based architecture, Inception and its variants [41–43] are unique such that they construct networks with convolutional layers using modules or blocks as opposed to building them. The modules of Inception are replaced with depthwise separable convolutions in Xception [44], an advanced extension of Inception.

## 3. MobileNetV2

A pre-trained version of the network called MobileNetV2 was loaded. This network uses the ImageNet database [45]. The pre-trained network is able to classify images into

different object categories. The modified MobileNetV2 model was used as a transfer learning model, and the Caffe model was used as a single shot detector (SSD) for identification and verification. The MobileNetV2 architectural classifier is an improved version of the MobileNetV1 architectural classifier. The MobileNetV2 model utilizes inverted residual blocks with linear bottlenecks over the MobileNetV1. It has a significantly lower parameter count than the original MobileNetV1. Larger image sizes provide a better performance, and MobileNetV2 supports any input size greater than $32 \times 32$. The block diagram for the MobileNetV2 architecture is represented in Figure 1. The model is like a CNN-based deep learning model that utilizes layers such as convolutional, pooling, dropout, non-linear, fully connected, and linear bottlenecks. The model consists of a $1 \times 1$ convolution layer, seventeen $3 \times 3$ convolutional layers, a max pooling average layer, and a classification layer. The algorithm must be run on sufficient datasets to train and classify the face with or without the mask. The aim of the proposed technique is to improve the accuracy of face mask detection with fewer trainable parameters. The most important building block in MobileNetV2 is known as the depth-wise separable convolution (DWSC) layer, which makes it very fast.

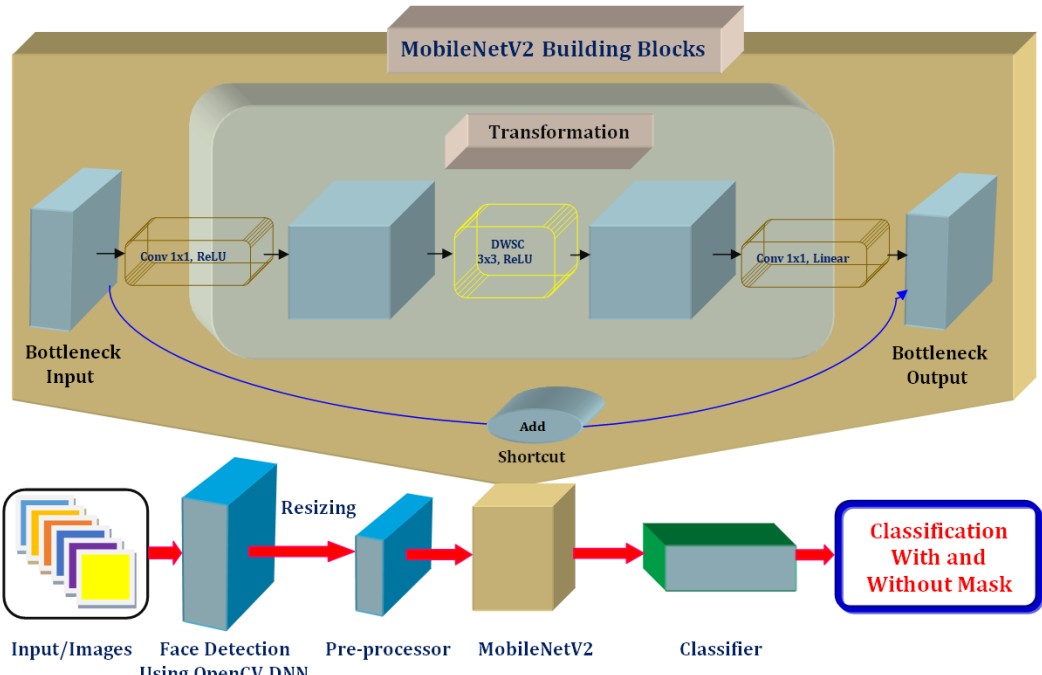

**Figure 1.** MobileNetV2 model.

The standard MobileNetV2 model was modified and improved by adding by our own five layers in the architecture, which includes average pooling of $7 \times 7$ size, flattening, a density of 128 neural networks, a dropout size of 0.5, and dense_1 of 2 for detection. Apart from these layers, the remaining layers were used for feature extraction. Each layer and individual building block, along with their output size, shape, and parameter count are explained in Tables 1 and 2. The modified MobileNetV2 model required 164,226 trainable parameters and 2,257,984 non-trainable parameters out of the 2,422,210 total parameters. The standard MobileNetV2 model, on the other hand, required 2,223,872 trainable parameters and 34,112 non-trainable parameters out of a total of 2,257,984 parameters. The main goal of the proposed model was to use a less number of trainable parameters for the improved MobileNetV2 model than the standard MobileNetV2 model, because it did not need to train the remaining parameters while they were being used for feature extraction for the detection of the face with and without a mask.

**Table 1.** Architecture of the Proposed Methodology-1(a).

| Head:Layers Name | Output Size | Parameters Used | | | |
|---|---|---|---|---|---|
| Input-1 (input layer) | (224,224,3) | 0 | | | |
| Convolution1D/Convolution2D | (112,112,32) | 864 | | | |
| Batch Normalization_Conv1 | (112,112,32) | 128 | | | |
| ReLU_Conv1 | (112,112,32) | 0 | | | |
| Extended Depthwise_Conv | (112,112,32) | 288 | | | |
| Extended Depthwise_Conv_BN | (112,112,32) | 128 | | | |
| Extended Depthwise_Conv_ReLU | (112,112,32) | 0 | | | |
| Extended Convolution2D | (112,112,16) | 512 | | | |
| Extended Batch Normalization_Conv | (112,112,16) | 64 | | | |
| **Block-1:Layers Name** | **Output size** | **Parameters used** | **Block-2:Layers Name** | **Output size** | **Parameters used** |
| Extended_Conv2D | (112,112,96) | 1536 | Extended_Conv2D | (56,56,144) | 3456 |
| Extended_BN | (112,112,96) | 384 | Extended_BN | (56,56,144) | 576 |
| Extended_ReLU | (112,112,96) | 0 | Extended_ReLU | (56,56,144) | 0 |
| Zero Padding2D | (113,113,96) | 0 | Depthwise_Convolution | (56,56,144) | 1296 |
| Depthwise_Convolution | (56,56,96) | 864 | Depthwise_BN | (56,56,144) | 576 |
| Depthwise_BN | (56,56,96) | 384 | Depthwise_ReLU | (56,56,144) | 0 |
| Depthwise_ReLU | (56,56,96) | 0 | Convolution2D | (56,56,24) | 3456 |
| Convolution2D | (56,56,24) | 2304 | Batch Normalization | (56,56,24) | 96 |
| Batch Normalization | (56,56,24) | 96 | Add | (56,56,24) | 0 |
| **Block-3:Layers Name** | **Output size** | **Parameters used** | **Block-4:Layers Name** | **Output size** | **Parameters used** |
| Extended_Conv2D | (56,56,144) | 3456 | Extended_Conv2D | (28,28,192) | 6144 |
| Extended_BN | (56,56,144) | 576 | Extended_BN | (28,28,192) | 768 |
| Extended_ReLU | (56,56,144) | 0 | Extended_Relu | (28,28,192) | 0 |
| Zero Padding2D | (57,57,144) | 0 | Depthwise_Conv | (28,28,192) | 1728 |
| Depthwise_Conv | (28,28,144) | 1296 | Depthwise_BN | (28,28,192) | 768 |
| Depthwise_BN | (28,28,144) | 576 | Depthwise_ReLU | (28,28,192) | 0 |
| Depthwise_ReLU | (28,28,144) | 0 | Convolution2D | (28,28,32) | 6144 |
| Convolution2D | (28,28,32) | 4608 | Batch Normalization | (28,28,32) | 128 |
| Batch Normalization | (28,28,32) | 128 | Add | (28,28,32) | 0 |
| **Block-5:Layers Name** | **Output size** | **Parameters used** | **Block-6:Layers Name** | **Output size** | **Parameters used** |
| Extended_Conv2D | (28,28,192) | 6144 | Extended_Conv2D | (28,28,192) | 6144 |
| Extended_BN | (28,28,192) | 768 | Extended_BN | (28,28,192) | 768 |
| Extended_ReLU | (28,28,192) | 0 | Extended_ReLU | (28,28,192) | 0 |
| Depthwise_Conv | (28,28,192) | 1728 | Zero Padding2D | (29,29,192) | 0 |
| Depthwise_BN | (28,28,192) | 768 | Depthwise_Conv | (14,14,192) | 1728 |
| Depthwise_ReLU | (28,28,192) | 0 | Depthwise_BN | (14,14,192) | 768 |
| Convolution2D | (28,28,32) | 6144 | Depthwise_ReLU | (14,14,192) | 0 |
| Batch Normalization | (28,28,32) | 128 | Convolution2D | (14,14,64) | 12,288 |
| Add | (28,28,32) | 0 | Batch Normalization | (14,14,64) | 256 |
| **Block-7:Layers Name** | **Output size** | **Parameters used** | **Block-8:Layers Name** | **Output size** | **Parameters used** |
| Extended_Conv2D | (14,14,384) | 24,576 | Extended_Conv2D | (14,14,384) | 24,576 |
| Extended_BN | (14,14,384) | 1536 | Extended_BN | (14,14,384) | 1536 |
| Extended_ReLU | (14,14,384) | 0 | Extended_ReLU | (14,14,384) | 0 |
| Depthwise_Conv | (14,14,384) | 3456 | Depthwise_Conv | (14,14,384) | 3456 |
| Depthwise_BN | (14,14,384) | 1536 | Depthwise_BN | (14,14,384) | 1536 |
| Depthwise_ReLU | (14,14,384) | 0 | Depthwise_ReLU | (14,14,384) | 0 |
| Convolution2D | (14,14,64) | 24,576 | Convolution2D | (14,14,64) | 24,576 |
| Batch Normalization | (14,14,64) | 256 | Batch Normalization | (14,14,64) | 256 |
| Add | (14,14,64) | 0 | Add | (14,14,64) | 0 |
| **Block-9:Layers Name** | **Output size** | **Parameters used** | **Block-10:Layers Name** | **Output size** | **Parameters used** |
| Extended_Conv2D | (14,14,384) | 24,576 | Extended_Conv2D | (14,14,384) | 24,576 |
| Extended_BN | (14,14,384) | 1536 | Extended_BN | (14,14,384) | 1536 |
| Extended_ReLU | (14,14,384) | 0 | Extended_ReLU | (14,14,384) | 0 |
| Depthwise_Conv | (14,14,384) | 3456 | Depthwise_Conv | (14,14,384) | 3456 |
| Depthwise_BN | (14,14,384) | 1536 | Depthwise_BN | (14,14,384) | 1536 |
| Depthwise_ReLU | (14,14,384) | 0 | Depthwise_ReLU | (14,14,384) | 0 |
| Convolution2D | (14,14,64) | 24,576 | Convolution2D | (14,14,96) | 36,864 |
| Batch Normalization | (14,14,64) | 256 | Batch Normalization | (14,14,96) | 384 |
| Add | (14,14,64) | 0 | | | |

**Table 2.** Architecture of the Proposed Methodology-1(b).

| Block-11:Layers Name | Output Size | Parameters Used | Block-12:Layers Name | Output Size | Parameters Used |
|---|---|---|---|---|---|
| Extended_Conv2D | (14,14,576) | 55,296 | Extended_Conv2D | (14,14,576) | 55,296 |
| Extended_BN | (14,14,576) | 2304 | Extended_BN | (14,14,576) | 2304 |
| Extended_ReLU | (14,14,576) | 0 | Extended_ReLU | (14,14,576) | 0 |
| Depthwise_Conv | (14,14,576) | 5184 | Depthwise_Conv | (14,14,576) | 5184 |
| Depthwise_BN | (14,14,576) | 2304 | Depthwise_BN | (14,14,576) | 2304 |
| Depthwise_ReLU | (14,14,576) | 0 | Depthwise_ReLU | (14,14,576) | 0 |
| Convolution2D | (14,14,96) | 55,296 | Convolution2D | (14,14,96) | 55,296 |
| Batch Normalization | (14,14,96) | 384 | Batch Normalization | (14,14,96) | 384 |
| Add | (14,14,96) | 0 | Add | (14,14,96) | 0 |
| **Block-13:Layers Name** | **Output size** | **Parameters used** | **Block-14:Layers Name** | **Output size** | **Parameters used** |
| Extended_Conv2D | (14,14,576) | 55,296 | Extended_Conv2D | (7,7,960) | 153,600 |
| Extended_BN | (14,14,576) | 2304 | Extended_BN | (7,7,960) | 3840 |
| Extended_ReLU | (14,14,576) | 0 | Extended_ReLU | (7,7,960) | 0 |
| Zero Padding2D | (15,15,576) | 0 | Depthwise_Conv | (7,7,960) | 8640 |
| Depthwise_Conv | (7,7,576) | 5184 | Depthwise_BN | (7,7,960) | 3840 |
| Depthwise_BN | (7,7,576) | 2304 | Depthwise_ReLU | (7,7,960) | 0 |
| Depthwise_ReLU | (7,7,576) | 0 | Convolution2D | (7,7,160) | 153,600 |
| Convolution2D | (7,7,160) | 92,160 | Batch Normalization | (7,7,160) | 640 |
| Batch Normalization | (7,7,160) | 640 | Add | (7,7,160) | 0 |
| **Block-15:Layers Name** | **Output size** | **Parameters used** | **Block-16:Layers Name** | **Output size** | **Parameters used** |
| Extended_Conv2D | (7,7,960) | 153,600 | Extended_Conv2D | (7,7,960) | 153,600 |
| Extended_BN | (7,7,960) | 3840 | Extended_BN | (7,7,960) | 3840 |
| Extended_ReLU | (7,7,960) | 0 | Extended_ReLU | (7,7,960) | 0 |
| Depthwise_Conv | (7,7,960) | 8640 | Depthwise_Conv | (7,7,960) | 8640 |
| Depthwise_BN | (7,7,960) | 3840 | Depthwise_BN | (7,7,960) | 3840 |
| Depthwise_ReLU | (7,7,960) | 0 | Depthwise_ReLU | (7,7,960) | 0 |
| Convolution2D | (7,7,160) | 153,600 | Convolution2D | (7,7,320) | 307,200 |
| Batch Normalization | (7,7,160) | 640 | Batch Normalization | (7,7,320) | 1280 |
| Add | (7,7,160) | 0 | | | |
| **Base:Layers Name** | **Output size** | **Parameters used** | | | |
| Convolution2D_Conv1 | (7,7,1280) | 409,600 | | | |
| Batch Normalization_Conv1 | (7,7,1280) | 5120 | | | |
| ReLU_Out | (7,7,1280) | 0 | | | |
| **AveragePooling2D** | **(1,1,1280)** | **0** | | | |
| **Flatten** | **(1280)** | **0** | | | |
| **Dense** | **(128)** | **163,968** | | | |
| **Dropout** | **(128)** | **0** | | | |
| **Dense_1** | **(2)** | **258** | | | |
| Total parameters: 2,422,210 | | | | | |
| **Trainable parameters: 164,226** | | | | | |
| Non-trainable parameters: 2,257,984 | | | | | |

The five additional layers used in the pretrained architecture of the MobileNetV2 model are highlighted in bold.

### 3.1. Depthwise Separable Convolution

A convolution is a mathematical technique that is commonly used by artificial neural networks (ANNs) to create a convolutional neural network (CNN). CNN can classify data as well as learn new characteristics by using image frames. CNNs are classified into several types, one of which is depthwise separable CNNs. This type of CNN is frequently used in comparison to traditional CNNs due to the following two factors: they have fewer parameters to change, which reduces overfitting. They are appropriate for mobile vision applications because they require fewer computations, making them less computationally expensive. This study examines the efficiency of depthwise separable convolutional networks compared with simple convolutional neural networks and describes the architecture and operations utilized in these networks.

The input data size was $D_F \times D_F \times M$, where $D_F \times D_F$ was the image size and $M$ was the number of channels (three for RGB images). A total of $N$ filters or kernels were considered for the convolutions of size $D_K \times D_K \times M$. The output size of a normal or standard convolution (SC) operation will be $D_P \times D_P \times N$, which is shown in Figure 2. The total complexity of the standard convolution can be computed by the multiplication of the size of the kernel which is equal to $D_K \times D_K \times M$ and the number of multiplications in one convolution operation. There were $N$ filters, and each filter traveled vertically and

horizontally $D_P$ times. Therefore, the total number of multiplications per convolution was $D_P \times D_P \times N$. The normal or standard operation cost or the complexity of convolution operations is given in the Equation (1).

$$SC_{cost} = (D_K \times D_K \times M)(D_P \times D_P \times N) = D_P{}^2 \times D_K{}^2 \times M \times N. \tag{1}$$

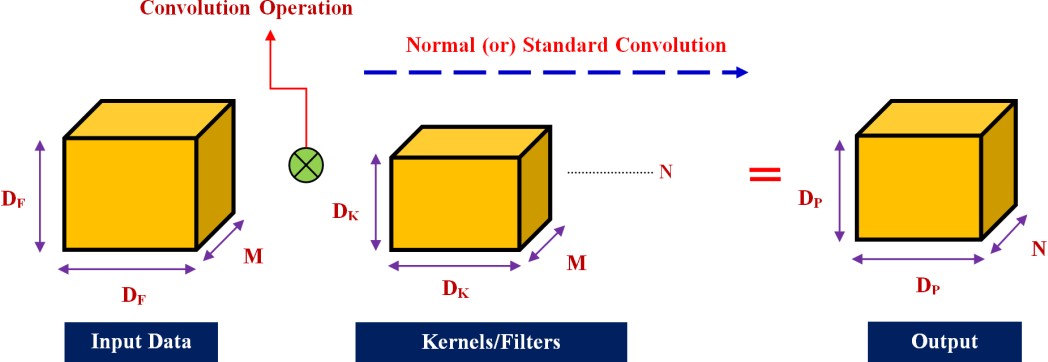

**Figure 2.** Standard Convolution.

Now coming to depthwise separable convolution (DWSC). This procedure is divided into two steps: depthwise convolution (DWC) and pointwise convolution (PWC). In contrast to standard convolution, which performs convolution on all $M$ channels at the same time, DWC operations only apply convolution to one channel at a time. As a result, the filters or kernels in this case will be $D_K \times D_K \times 1$. It follows that $M$ of these filters are needed because there are $M$ channels in the input data. Therefore, the output will be obtained as $D_P \times D_P \times M$ as shown in Figure 3. $D_K \times D_K$ multiplications are required for a single convolution operation, since the filter is multiplied by $D_P \times D_P$ across each of the $M$ channels. Equation (2) gives the depthwise convolution operation cost.

$$DWC_{cost} = (D_K \times D_K \times 1)(D_P \times D_P \times M) = D_P{}^2 \times D_K{}^2 \times M. \tag{2}$$

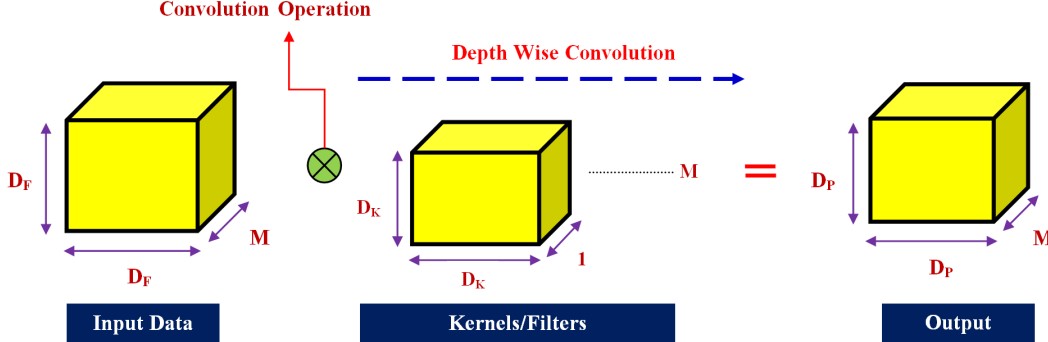

**Figure 3.** Depthwise Convolution.

In the pointwise operation, $1 \times 1$ convolution is performed on the M channels. This procedure will employ a $1 \times 1 \times M$ filter. The output size changes to $D_P \times D_P \times N$ when $N$ such filters are obtained as shown in Figure 4. Each convolution operation necessitates $1 \times M$ multiplications. Since $D_P \times D_P$ times travel through the filter. The pointwise convolution operation cost is given in Equation (3).

$$PWC_{cost} = (1 \times 1 \times M)(D_P \times D_P \times N) = D_P{}^2 \times M \times N. \tag{3}$$

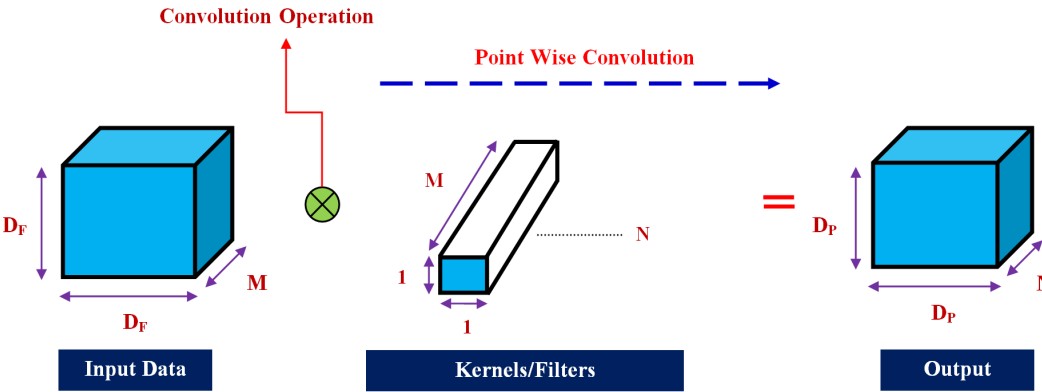

**Figure 4.** Pointwise Convolution.

The overall depthwise separable convolution (DWSC) cost is computed by sum of the cost of depthwise convolution and the cost of pointwise convolution. The computational cost of convolution operations is given in Equation (4).

$$DWSC_{cost} = (D_P{}^2 \times D_K{}^2 \times M) + (D_P{}^2 \times M \times N) = (D_P{}^2 \times M)(D_K{}^2 + N). \quad (4)$$

The proposed methodology MobileNetV2 utilized the depthwise separable convolution, which is well known for its fast computational time, and lower computational cost compared to other models. The reduced computation cost of DWSC can be computed by taking the ratio of the DWSC cost ($DWSC_{cost}$) and the SC cost ($SC_{cost}$), which is given below in Equation (5).

$$Ratio(R) = \frac{DWSC_{cost}}{SC_{cost}} = \frac{(D_P{}^2 \times M) \times (D_K{}^2 + N)}{D_P{}^2 \times D_K{}^2 \times M \times N} = \frac{1}{N} + \frac{1}{D_K{}^2}. \quad (5)$$

For $N = 224$ and $D_K = 3$, the DWSC cost is reduced by nine times as compared to the SC cost.

### 3.2. Inverted Residual Blocks

The inverted residual blocks consist of a skip connection connecting the beginning and end of a convolutional block. The network will be able to access previous activations that were not changed by the convolutional block by including these two states, which helps to create networks with a lot of depth. A detailed observation of the skip connection shows that an original residual block uses a wide-narrow-wide strategy. The input contains many channels, which are reduced via a low-cost $1 \times 1$ convolution. The next $3 \times 3$ convolution will then have fewer parameters as a result. In the end, another $1 \times 1$ convolution is used to expand the number of channels to add input and output. On the other hand, the MobileNetV2 utilizes inverted residual blocks as a narrow-wide-narrow strategy as shown in Figure 5. In order to reduce the number of parameters for the subsequent $3 \times 3$ depthwise convolution, the network is first made wider using a $1 \times 1$ convolution. The network is then restricted by another $1 \times 1$ convolution to match the original number of channels.

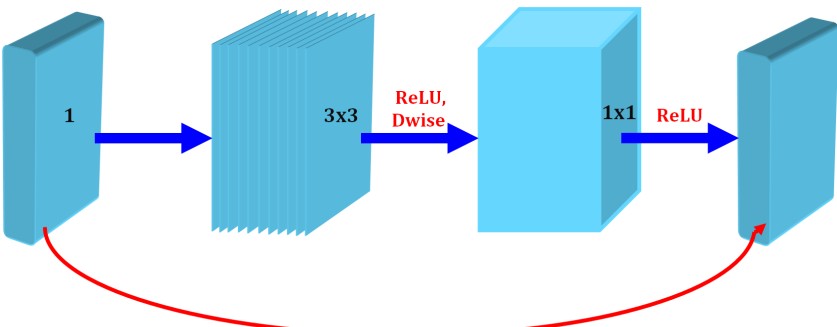

**Figure 5.** Inverted Residual Block.

### 3.3. Linear Bottlenecks

A linear bottleneck is an idea in which the final convolution of a residual block has a linear output before it is added to the initial activations. The compressed, low-dimensional representation is fed into the module after it has been quickly depthwise convoluted into higher dimensionality before being filtered. Using a linear convolution, the features contribute to a low-dimensional form. The collection of layer activation functions for a set of real images used as input is the region of interest. According to this intuition, the overall dimension of the activation space can be reduced using the width multiplier methodology until the region of interest occupies the entire area.

The experimental results show that utilizing linear layers seems important because it protects against non-linearity by removing excessive information. Utilizing shortcuts straight in between bottlenecks is preferred because of the belief that they truly include all the information required, even though such expansion layers are more simply associated with an implementation detail for one non-linear tensor modification. According to the purpose of traditional residual connections, the idea of adding shortcuts is really to increase one's capacity towards such a gradient that will spread over numerous levels. The tests show that the inverted architecture performs much better and uses a less memory. It is a key element of the architecture that input or output areas of the construction blocks such as bottleneck layers and modification of layers, a non-linearity functional which changes the input to output, will be properly separated.

### 4. Materials and Methods

The proposed methodology includes the CMNV2 model, which is used for photo images and real-time face mask detection on live webcam video streaming. The proposed CMNV2 method was created with the DNN architecture using TensorFlow, Keras, and OpenCV libraries to detect face masks in real-time. In the proposed model, two sets of files were used to explore the model architecture and store the weights for each layer. The first is the prototxt file, which contains the layer model architecture, and the second is the Caffe model, which stores the weights for each layer. An image may contain multiple objects, with non-linear boundaries between them. An activation function known as ReLU was used after each convolution layer to introduce non-linearity into the neural network. The ReLU function is linear for all positive values of the convolution layer output and zeroes for all negative values. After training the proposed model, this model was used for the classification of photos and real-time video images, as shown in the block diagram of Figure 6.

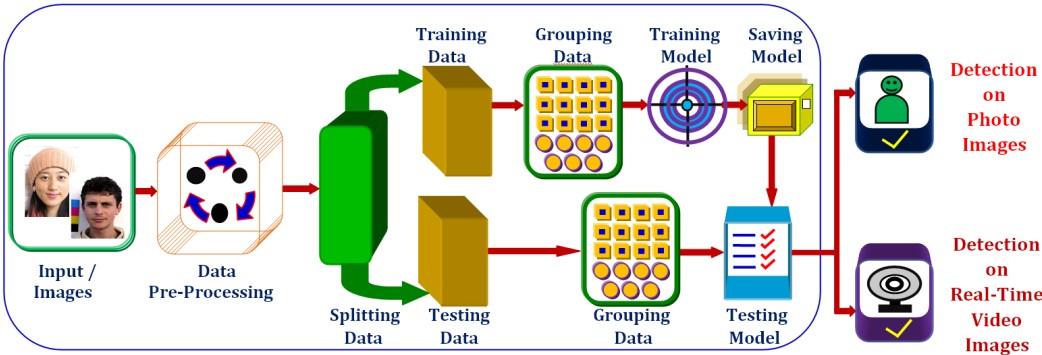

**Figure 6.** Block Diagram.

The following steps describe the above block diagram of the proposed CMNV2 model.

Step 1: In this work, a dataset of the joint photographic expert group (JPEG)/ portable network graphics (PNG) images of people's faces with and without masks was used.

Step 2: The images were arranged in two folders, one with a mask and one without a mask. The images were then labeled using the graphical tool 'labeling'. Both the training and testing datasets for these labels contained label data for each image.

Step 3: The data of a total of 1376 images were randomly split into training and testing each time, and stratification was used to determine the ratio of masked and unmasked data in the training and test data.

Step 4: The collected images were mainly split into two categories. The first one of them is a training dataset consisting of 1100 images, representing 80% of the entire dataset. The second category is the testing data, which consists of the remaining 276 images, representing 20% of the total dataset.

Step 5: After processing the training and testing data, various batches were used to determine the number of samples to process before updating the model.

Step 6: The model training was used to run the algorithm on the input data and compare the processed results with the sample output, which is explained in Section 4.2.

Step 7: After loading the modified MobileNetV2 and Caffe model, the proposed CMNV2 model was saved and later utilized for the testing dataset as explained in Section 4.3.

Step 8: Testing the model output results with the display of photo and real-time videos images is explained in Section 4.4.

Prajna Bhandary created a dataset of 1376 images related to the artificial mask by adding facial masks to standard images of faces to categorize the class of masked images dataset. There are 690 images with masks and 686 images without masks among them. The 'Prajna Bhandary' dataset, which is available and taken from GitHub [46], contains images of people with and without masks. During the proposed methodology, we used an additional five of our own images, one with a mask and four without, to test the model's accuracy on the different datasets. The proposed research paper has a total data of 1381 images as shown in Figure 7.

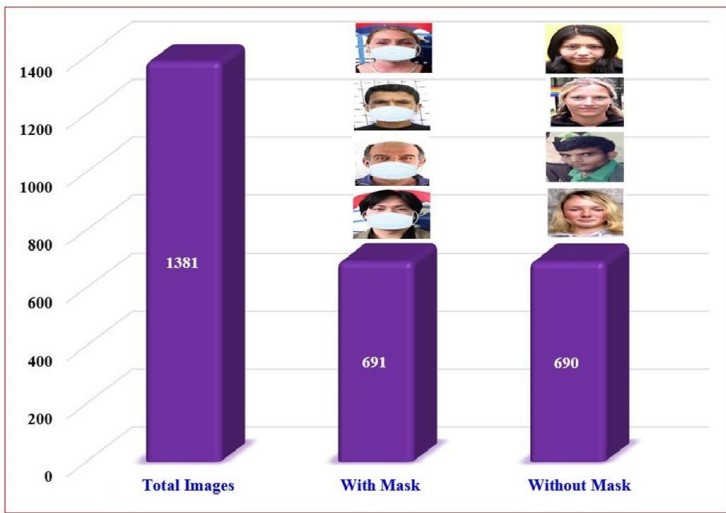

**Figure 7.** Data Visualization.

### 4.1. Data Preprocessing

The dataset was saved in a folder on the laptop after being downloaded so that the following step may be accessed from there. Pre-processing is a function that loads all files from a folder as input and resizes the images for future use with the proposed model. The images are pre-processed by adding the data after loading and converting it to an array. To speed up the calculation, the images were converted to a numerical python (NumPy) library, utilized for working with the values. The pattern was first analyzed by storing the image data in the label binarizer (LB) to identify categories 0 and 1 integer values. The next step was to go to the images and split them into training and testing sets.

### 4.2. Model Training

A detailed representation of the proposed model design flow is shown in Figure 8. As a base model, the pre-trained MobileNetV2 model was loaded from ImageNet. After loading the base model, we added our own five layers for the transfer learning, as shown in Table 2. The added layers consist of average pooling of $7 \times 7$ size, flattening, a density of 128 neural networks, a dropout size of 0.5, and dense_1 of 2. These layers helped to improve the performance of the modified MobileNetV2 using fewer parameters by obtaining an average value using the average pooling layer. Flattening helps to come into a single dimension. A dense layer such as an activation function such as rectified linear unit (ReLU) was used to create fully connected layers, in which every output depends on every input. Certainly, preventing overfitting is helped by adding a dropout layer, followed by adding two dense layers for binary classification. Due to the existence of more than one output neuron, the softmax function was utilized.

To prevent the loss of previously acquired characteristics, the base layers were frozen. A new set of trainable layers was added, and these layers were trained using the acquired dataset to find the attributes that could be utilized to classify a mask-wearing face from a face without a mask. Following that, the weights were saved and the model was modified. By using these pre-trained models, a model can take the benefit of bias weights without spending extra computational expenses and the ability to retain learned features without forgetting them. Afterward, the summary of the proposed methodology is included. Then, the modified MobileNetV2 model was trained with more additional parameters such as a learning rate of 0.001, 35 epochs and a batch size of 21. Many deep learning applications, such as computer vision, use the Adam optimizer. We utilized the Adam optimizer, which adjusts the learning rate for each neural network weight by estimating the first and second moments of the gradient. The suitable algorithm performed transfer learning to modify the weights in the neural network after each batch was received. A neural network was trained

and finally saved for use in making predictions on the image dataset and the real-time video images.

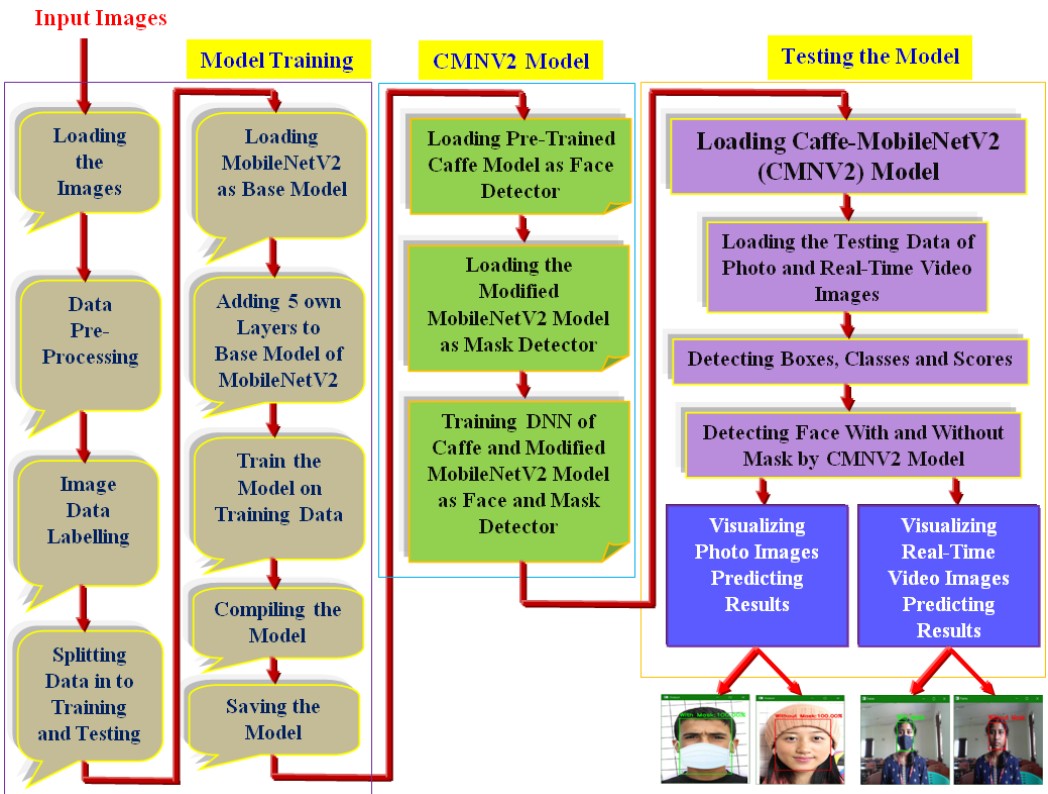

**Figure 8.** Flow of the Design.

### 4.3. CMNV2 Model

The Caffe-MobileNetV2 (CMNV2) model is the proposed model that combines Caffe and MobileNetV2 for face mask detection. After training the CMNV2 model, the model can be used for face mask detection as the Caffe model for FaceNet and the modified MobileNetV2 model for MaskNet. Loading a face detector model which is better designed as a Caffe model used a pre-trained model which needed to import two files of prototxt as well as loading the modified MobileNetV2 model as a mask detector. The DNN was further trained to detect and predict using the modified MobileNetV2 model for MaskNet, along with the Caffe model for FaceNet, and then the frame's dimensions were captured and a blob function was created. The blob function's input and initialization of the list of faces, their corresponding locations and list of predictions were set. The DNN model was used to process the previously imported files and also loaded the modified MobileNetV2 main model by adding parameters. When the test images were loaded, the model took the loaded image and converted it to a value for the classification.

### 4.4. Model Testing

After loading the Caffe-MobileNetV2 (CMNV2) model, the testing data of photos and real-time video images were loaded for determining whether the person is wearing a mask by predicting their facial features and obtaining a level of confidence. Although there is a probability of detection on photo images, it will first compute the X and Y coordinates of the bounding box of the objects before detecting the face if the probability is greater than the threshold value for real-time video image detection. It was ensured that the bounding boxes fit within the frame's boundaries by establishing the class label 'with mask' in green and 'without mask' in red. These were used to create the text and bounding box. The text contained the label with a probability value for photo image detection and should contain the label for real-time video detection. Images were scaled with RGB dimensions

of photo images and real-time video images in different channels using the blob function by utilizing a combination of prototxt files; the face was detected by adding the label and bounding boxes. Finally, the prediction of the photo images and real-time video images output results are displayed.

## 5. Performance Discussion of the Proposed CMNV2 Model

As discussed in Section 4, a total of 1376 images created by Prajna Bhandary were used in this research paper. The dataset was divided into 1100 images that represent 80% of the dataset and 276 images that represent 20% of the dataset. A total of 1100 images were used to train the proposed model and 276 images were used to test the model. The test dataset consists of 138 images with masks and the remaining 138 images without masks. After training of the model, the test accuracy of the proposed model is 99.64% for face mask classification.

The performance measures of the proposed CMNV2 model are also presented using a confusion matrix, which is displayed in the form of a summarized table and is used to assess the efficiency of the model's classifications [47]. A true positive occurs when the predicted class and the actual class of the data point are both 'with mask'. A true negative occurs when the predicted class is 'without mask' and the actual class of the data point is also 'without mask'. False positives occur when the actual class of a data point is 'without mask' but the predicted class is 'with mask'. False negatives occur when the actual class of a data point is 'with mask' but the predicted class is 'without mask'.

The generalized confusion matrices and the confusion matrix for the classification by proposed model are shown in Figure 9a,b, respectively. The proposed CMNV2 methodology was evaluated based on the acquired measures, such as accuracy, precision, recall (or) sensitivity, f1-score and error rate. From Figure 9b it can be clearly observed that the true positive (TP) value is 137, the true negative (TN) value is 138, the false positive (FP) value is 0, and the false negative (FN) value is 1.

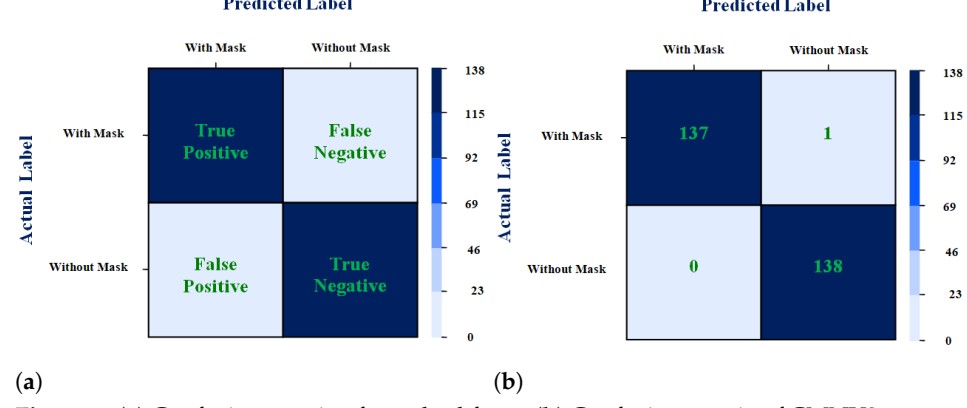

(**a**)                                                                                                          (**b**)

**Figure 9.** (**a**) Confusion matrix of standard form, (**b**) Confusion matrix of CMNV2 proposed model.

For the proposed CMNV2 model using a two-class binary classifier, metrics for the output of binary classification with and without masks were developed and the accuracy, precision, recall, f1-score, and error rate were obtained.

### 5.1. Accuracy

The accuracy is the most widely used evaluation metric for recognition and categorization challenges. It calculates the proportion of correct predictions to the total number of samples. The Equation (6) gives the model accuracy for classification.

$$Accuracy = \frac{(TP + TN)}{(TP + FP + FN + TN)} = 0.9964 = 99.64\%. \tag{6}$$

*5.2. Precision*

Precision in positive observations is defined as the ratio of correctly predicted positive observations to all predicted positive observations. Equation (7) gives the classification precision value.

$$Precision = \frac{TP}{(TP + FP)} = 1 = 100\%. \tag{7}$$

*5.3. Recall*

The recall is defined as the ratio of correctly predicted positive observations to all actual class observations. Equation (8) gives the recall value for the classification.

$$Recall = \frac{TP}{(TP + FN)} = 0.9928 = 99.28\%. \tag{8}$$

*5.4. F1-Score*

The *F1-score* is computed by harmonic mean of precision and recall. As a result, this score considers both false positives and false negatives. Equation (9) yields the classification's *F1-score*.

$$F1 - score = \frac{(2 \times Precision \times Recall)}{(Precision + Recall)} = 0.9964 = 99.64\%. \tag{9}$$

*5.5. Error Rate*

The reverse of the accuracy is the error rate, also known as the misclassification rate. This metric indicates the number of samples from both the positive and negative classes that are misclassified. Equation (10) gives the error rate of the model for classification.

$$Error rate = (1 - Accuracy) = 0.0036 = 0.36\%. \tag{10}$$

The developed CMNV2 model can detect faces with and without masks accurately and uses fewer parameters than the existing models, according to experiments.

According to our research paper, the overall model achieved an accuracy of 99.64% and when it comes to precision calculations, the false positive (FP) value is zero. The 100% precision value is obtained by substituting the values of the confusion matrix into Equation (7) in our manuscript.

## 6. Results Visualization

All of the research was completed on a laptop with an Intel(R) with Core(TM) of i3-1005G1 CPU processor with 1.2 GHz, 8 GB of RAM, and a 64-bit operating system with an x64-established processor. For the construction and implementation of the many experimental paths in the proposed paper, Python 3.9 version kernel, and Jupyter Notebook tools, were utilized. The proposed CMNV2 model was applied to the photo image dataset and to the extracted real-time video images from the webcam for the classifications.

The proposed model makes predictions based on the training dataset's pattern and labels. The visualization of the classification results of the photo images data and video images are shown in Figure 10 and Figure 11, respectively. The face with a proper mask is displayed with a green bounding box and the face without/with an incorrect mask is displayed with a red bounding box. The classification accuracy of our model is 99.64% on the test dataset. The accuracy and the loss curve for the training and testing data of the images are shown in Figure 12. The training and test accuracy are represented by the green colour and blue colour, respectively, in Figure 12a. The training and testing loss are represented by the green colour and blue colour, respectively, in Figure 12b. Table 3 indicates that the proposed model produces better results in all metrics for the classification of faces with and without masks.

The proposed CMNV2 model was mainly tested and trained on images of straight faces with and without masks. This model performs less accurately in real-time video images with sudden facial movements than in still images.

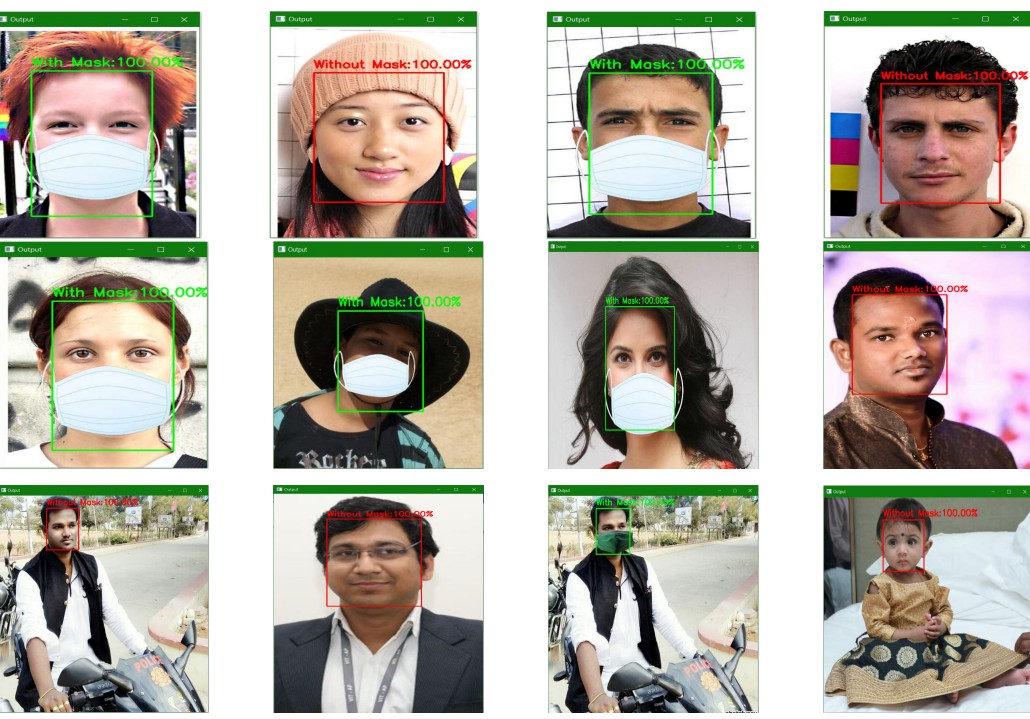

**Figure 10.** Face detection with and without masks in photo images.

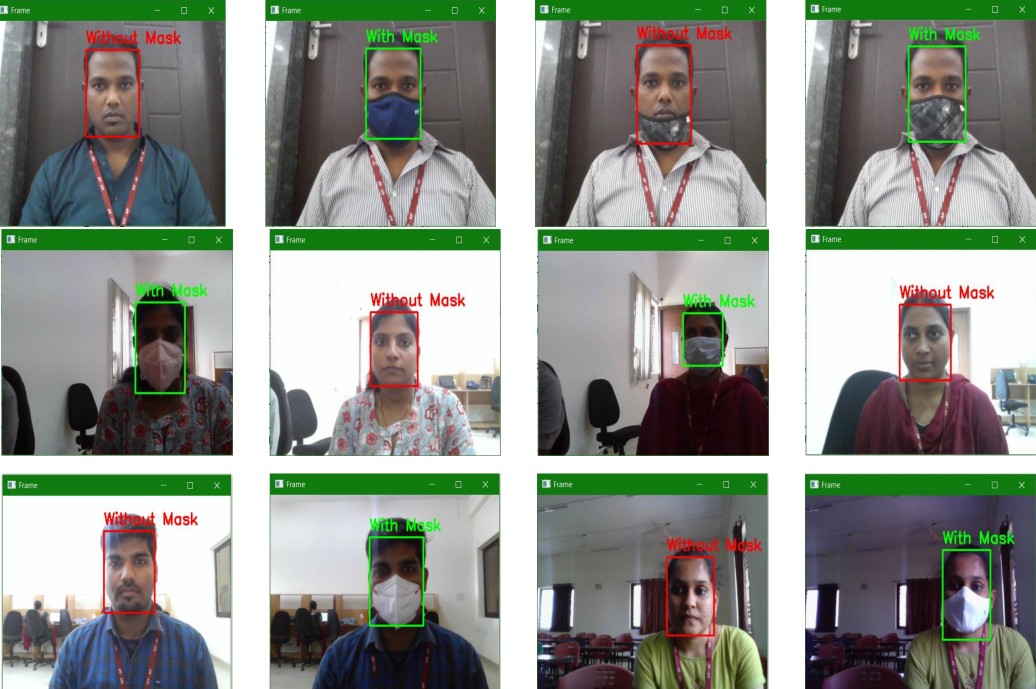

**Figure 11.** Face detection with and without masks in real-time video images.

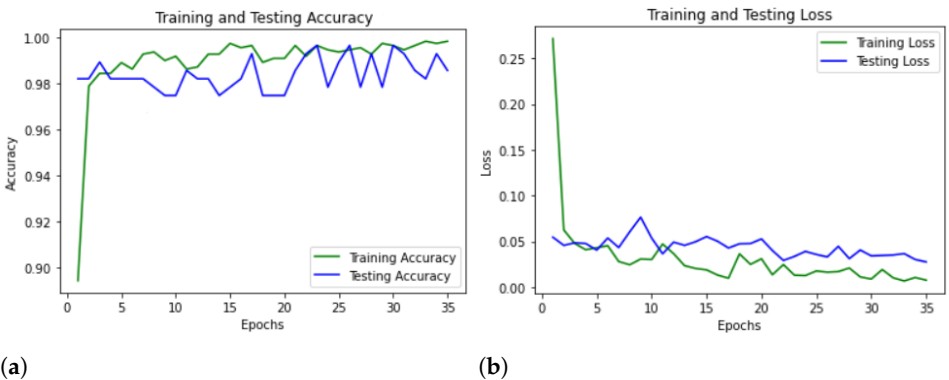

**Figure 12.** (**a**) Accuracy curve for training and testing, (**b**) Loss curve for training and testing.

**Table 3.** Comparison of different dimension images of MobileNetV1 and MobileNetV2 implemented along with Caffe model.

| Sl.No. | Model Version (Dimension) | Accuracy% | Precision% | Recall% | F1-Score% | Error Rate% |
|---|---|---|---|---|---|---|
| 1 | MobileNetV1 (128 × 128) | 93.84% | 99.18% | 88.40% | 93.48% | 6.16% |
| 2 | MobileNetV1 (160 × 160) | 94.20% | 99.19% | 89.13% | 93.89% | 5.80% |
| 3 | MobileNetV1 (192 × 192) | 94.56% | 99.20% | 89.85% | 94.29% | 5.44% |
| 4 | MobileNetV1 (224 × 224) | 94.92% | 98.43% | 91.30% | 94.73% | 5.08% |
| 5 | MobileNetV2 (128 × 128) | 97.82% | 98.53% | 97.10% | 97.80% | 2.18% |
| 6 | MobileNetV2 (160 × 160) | 98.19% | 98.54% | 97.82% | 98.18% | 1.81% |
| 7 | MobileNetV2 (192 × 192) | 98.55% | 98.55% | 98.55% | 98.55% | 1.45% |
| 8 | MobileNetV2 (224 × 224) | 99.64% | 100% | 99.28% | 99.64% | 0.36% |

The classes are evenly distributed, and accuracy calculated using Equation (6) is a good starting point. The performance measure of the several versions of MobileNetV1 and MobileNetV2 with different dimension images are also calculated and compared in terms of various metrics such as accuracy, precision, recall, f1-score, and error rate as shown in Table 3 and displayed in Figure 13a–c.

The results of our proposed CMNV2 model in terms of accuracy and with fewer parameters utilized in comparison with other various models are summarised in Tables 4 and 5, which show that the proposed CMNV2 methodology performed better than other existing models.

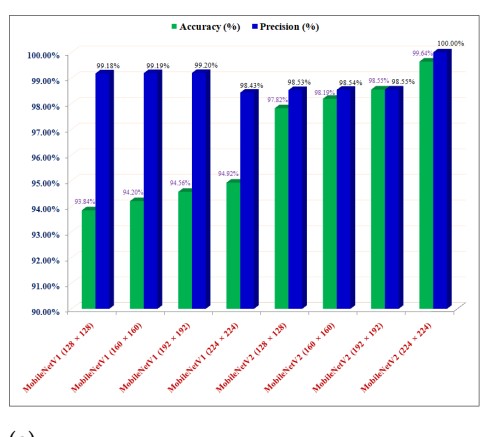

(**a**)

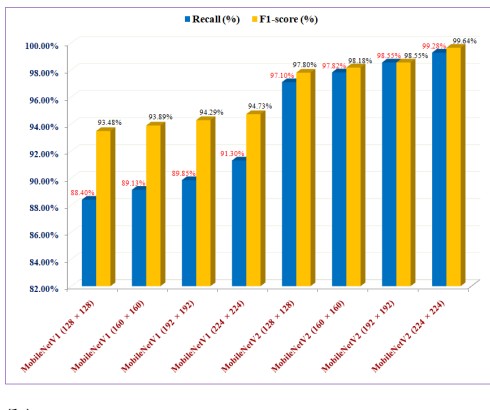

(**b**)

**Figure 13.** *Cont.*

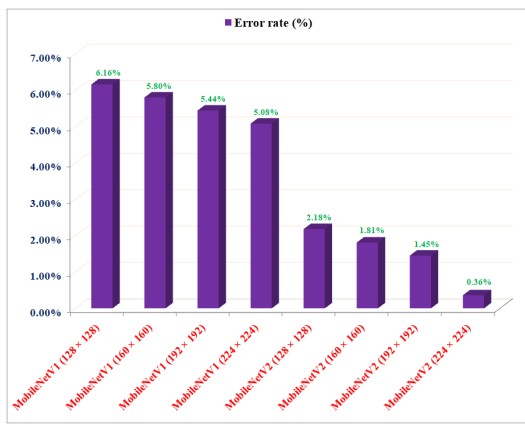

(**c**)

**Figure 13.** Comparison graphs of various dimension images: (**a**) accuracy, precision (**b**) recall and f1-score and (**c**) error rate of MobileNetV1 and MobileNetV2 implemented along with Caffe model.

**Table 4.** Comparison of various models' accuracy.

| Sl.No. | Model Name | Year | Accuracy% |
|---|---|---|---|
| 1 | ResNet50 [19] | 2021 | 47.00% |
| 2 | OpenFace [16] | 2020 | 63.18% |
| 3 | DeepFace [16] | 2020 | 63.78% |
| 4 | MTCNN+FaceNet [48] | 2020 | 64.23% |
| 5 | FaceNet [16] | 2020 | 67.48% |
| 6 | VGG-Face [16] | 2020 | 68.17% |
| 7 | IAMGAN [28] | 2020 | 86.50% |
| 8 | FaceMaskNet21 [29] | 2022 | 88.92% |
| 9 | DSA-Face [31] | 2021 | 91.20% |
| 10 | CNNs [20] | 2022 | 91.30% |
| 11 | CBAM [30] | 2021 | 92.61% |
| 12 | SSDMNV2 [32] | 2021 | 92.64% |
| 13 | GANs [33] | 2020 | 94.10% |
| 14 | MGL [34] | 2020 | 95.00% |
| 15 | FaceNet [49] | 2020 | 97.00% |
| 16 | LPD [35] | 2020 | 97.94% |
| 17 | CNN [36] | 2022 | 98.00% |
| 18 | ResNet50 [37] | 2021 | 98.20% |
| 19 | LW-CNN [50] | 2022 | 98.47% |
| 20 | CMNV2 (Proposed Model) | 2022 | 99.64% |

Our CMNV2 model in terms of accuracy is represented in Table 4 and displayed in Figure 14. Our proposed model of CMNV2 requires fewer parameters for face mask detection compared to other different methodologies [50,51] as illustrated in Table 5.

The 224 × 224 photo image dimension that was utilized in this research presents higher classification accuracy as compared to 192 × 192, 160 × 160, and 128 × 128 size image dimensions. This has been theoretically and practically seen throughout testing on the photo and real-time video images. This proposed method can be integrated with CCTV cameras, and the data can be used to determine whether any of their employees or patients are not wearing masks in hospitals or offices. If a person is discovered without a mask, a message can be sent to instruct them to wear one. This methodology can be helpful to maintain safety requirements in an attempt to prevent the spread of any airborne disease.

Limitations of the study: The proposed CMNV2 model was mainly tested and trained on images of straight faces with and without masks. Our model is less accurate with sudden facial movements in real-time video images, than it is with still faces. The overall model accuracy of our research decreased due to this.

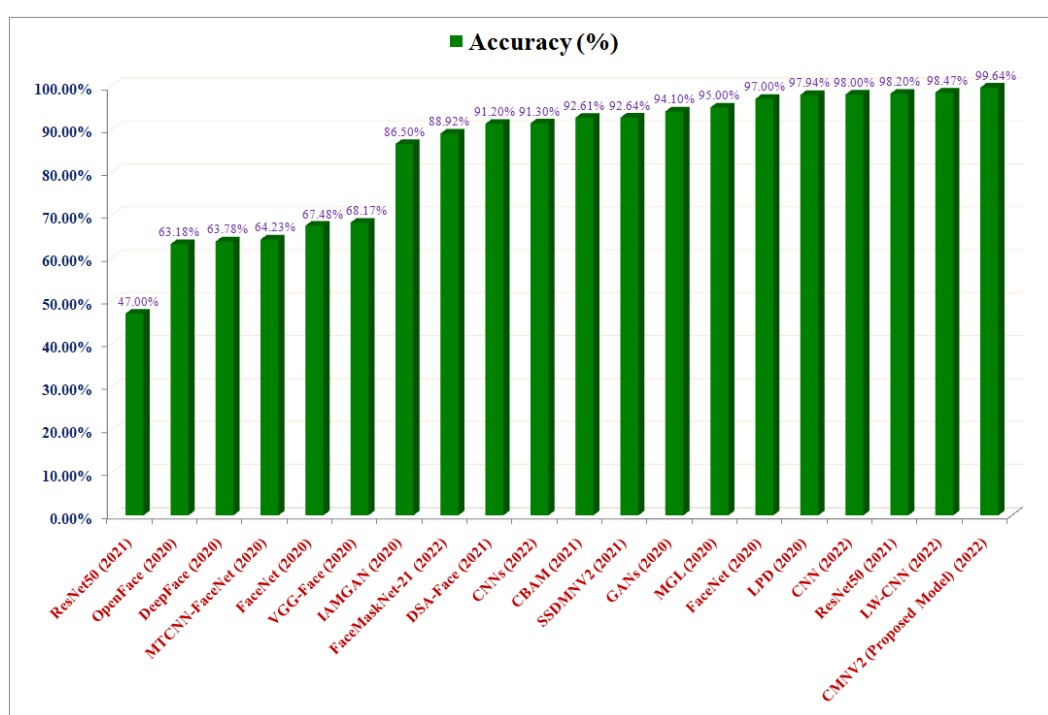

**Figure 14.** Comparison graph of various models' accuracy.

**Table 5.** Comparison of various models' parameters and total layers.

| Sl.No. | Name of the Model | Parameters Used | Total Layers Used |
| --- | --- | --- | --- |
| 1 | VGG19 | 143,000,000 | 19 |
| 2 | VGG16 | 138,000,000 | 16 |
| 3 | AlexNet | 62,000,000 | 8 |
| 4 | InceptionV2 | 56,000,000 | 48 |
| 5 | Inception-ResNetV2 | 56,000,000 | 164 |
| 6 | ResNet101 | 44,000,000 | 101 |
| 7 | InceptionV4 | 43,000,000 | 164 |
| 8 | InceptionV3 | 24,000,000 | 48 |
| 9 | Xception | 23,000,000 | 71 |
| 10 | ResNet50 | 22,500,000 | 50 |
| 11 | MobileNetV1 | 13,000,000 | 30 |
| 12 | GoogleNet | 7,000,000 | 27 |
| 13 | MobileNetV2 | 3,500,000 | 53 |
| 14 | CMNV2 (Proposed Model) | 164,226 | 159 |

## 7. Conclusions

The proposed research describes the methodology for detecting masked faces by modifying the MobileNetV2 method and the Caffe model. Our research implemented the proposed model with software using Python 3.9 related libraries such as TensorFlow, Pandas, Keras, scikit learn (sklearn) and OpenCV. The installed Anaconda Navigator experience of Juypter notebook made it easy and convenient to continue our research, compared to using Pytorch. The transfer learning technique helped the model to distinguish between people wearing or not wearing a face mask. The dataset of images and real-time video frames from a webcam were used to validate the efficiency of the face mask classifier. The effectiveness of the model was determined using accuracy, precision, recall, f1-score, and error rate. The proposed methodology worked substantially very well on the dataset with an accuracy of 99.64%. A face mask classifier was also used for real-time video images. Experimental results for video images are accurate when faces are localized by a webcam using the CMNV2 model after the classifier detects the face locations. Using

video surveillance as an input in public places, government agencies can benefit from this detection process. An implementation of this approach was successfully tested in real-time by installing a model of CMNV2. The proposed method can significantly reduce violations through real-time intervention, thus saving time and improving public safety by slowing airborne transmission. A facial mask detection approach in real-time can be used in a variety of places such as in airports, shopping malls, subway stations, temples, offices, hospitals, educational institutions, etc.

**Author Contributions:** Conceptualization, B.A.K.: methodology, software, writing, editing, original draft preparation, Conceptualization; M.B.: validation, formal analysis, investigation, resources, writing review and editing, visualization, supervision, project administration. All authors have read and agreed to the published version of the manuscript.

**Funding:** This research received no external funding.

**Institutional Review Board Statement:** Not applicable.

**Informed Consent Statement:** Not applicable.

**Data Availability Statement:** Data of written code, all results of photo and real-time video images will be made available on request.

**Conflicts of Interest:** The authors declare no conflict of interest.

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
