# Peer review of "Face Mask Detection on Photo and Real-Time Video Images Using Caffe-MobileNetV2 Transfer Learning"

_applsci, doi:10.3390/app13020935_

Round 1

Reviewer 1 Report

The paper provides interesting study related to face mask detection on photo and real-time video images using caffe-mobile Netv2 transfer learning

This manuscript needs a lot of improvement.

 Abstract:

Add something about the benefits results of the research. 

Introduction

-Improve literature review…Add other related studies, citations should be above 50.

-Can you please establish the research gap in simple lines at the end of intro. [Please add]

Materials and Methods:

-Fig.6.. write this section in stepwise pattern also.

Step.1:

Step:2

Step:3…etc.

Discussion:

This section must contain implications for research, practice and/or Field: Does the paper identify clearly any implications for research, practice and/or society? Does the paper bridge the gap between theory and practice? How can the research be used in practice (economic and commercial impact), to influence technical policy, in research (contributing to the body of knowledge)? Add something for field professionals. [Please add]

Limitations of the study:

Please add as heading about the limitations of the study.

Reviewer 2 Report

The proposed method is very well explained.

There are some spelling and language errors.

Conclusions section is a bit weak compared to this article. It needs to be developed.

How can Table 1 be explained more simply? This is just a suggestion. Normally, it is understood from the table, but for easier understanding.

The shapes in the article are beautiful, but softer colors can be preferred. (It is only a suggestion. There is no scientific deficiency.)

The following article should be included in the related works section.

https://doi.org/10.1016/j.bspc.2021.103216

Reviewer 3 Report

Review Comments:

Journal: Applied Sciences (ISSN 2076-3417)

Date of Review: Dec 12, 2022

Face Mask Detection on Photo and Real-Time Video Images using Caffe-MobileNetV2 Transfer Learning

This technical article, Face Mask Detection on Photo and Real-Time Video Images using Caffe-MobileNetV2 Transfer Learning, is novel research focussing on Face detection systems of people from the photo and in real-time video images with and without a mask. This idea is implemented based on the features around their eyes, ears, nose, and forehead by using the original masked and unmasked images to form a baseline for face detection. The machine learning library used are Caffe-MobileNetV2 (CMNV2) model for feature extraction and image classification. The Caffe model is implemented as a face detector, and the MobileNetV2 model is used as a mask detector. In this work, five different layers are added to the pre-trained MobileNetV2 architecture for better classification accuracy with fewer training parameters to the given data for face mask detection.

  1. Issue #1 The formulas mentioned in the paper for accuracy, precision, recall, and F1-score are pretty straightforward; do we need to include them in the manuscript? 

  2. Fig 10 talks about the success rate. Did you encounter somewhere the trained model would not be able to mask properly? 

  3. Did the researchers explore PyTorch and Tensorflow, OpenAI, or CNTK and get a comparative analysis of the results? CVzone is another avenue that could be used for analyzing images. 

  4. Fig 11 a. Mentions 100% accuracy in the last iteration, I am not sure if a model could be trained with such accuracy that it starts giving 100% accuracy. 

  5. The paper is OK, except a few sentences require rephrasing.

  6. The paper will be accepted after addressing the suggestive changes.

Round 2

Reviewer 1 Report

This paper can be accepted in present form.

Author Response

Dear Reviewer,

We appreciate the reviewer for taking the time and effort to provide your valuable feedback on our manuscript. Thank you for acceptance our work for the publication.

Reviewer 3 Report

I couldn't see the document where changes have been incorporated. It's hard to follow in the attached PDF. 

Author Response

We appreciate the reviewer's time and effort in providing valuable feedback on our manuscript. In our revised manuscript, we clarify by changing the text color of the rebuttal report to the same color.
